# A Novel Active Cooling System for Internal Combustion Engine Using Shape Memory Alloy Based Thermostat

**DOI:** 10.3390/s23083972

**Published:** 2023-04-13

**Authors:** Pacifique Turabimana, Jung Woo Sohn, Seung-Bok Choi

**Affiliations:** 1Department of Aeronautics, Mechanical and Electronic Convergence Engineering, Graduate School, Kumoh National Institute of Technology, Gumi 39177, Republic of Korea; 2Department of Mechanical Design Engineering, Kumoh National Institute of Technology, Daehak-Ro 61, Gumi 39177, Republic of Korea; 3Department of Mechanical Engineering, The State University of New York, Korea (SUNY Korea), Songdo Moonhwa-Ro, Incheon 21985, Republic of Korea; 4Department of Mechanical Engineering, Industrial University of Ho Chi Minh City (IUH), 12 Nguyen Van Bao Street, Ho Chi Minh City 70000, Vietnam

**Keywords:** shape memory alloy, active cooling system, thermostat, internal combustion engine

## Abstract

Pollutants in exhaust gases and the high fuel consumption of internal combustion engines remain key issues in the automotive industry despite the emergence of electric vehicles. Engine overheating is a major cause of these problems. Traditionally, engine overheating was solved using electric pumps and cooling fans with electrically operated thermostats. This method can be applied using active cooling systems that are currently available on the market. However, the performance of this method is undermined by its delayed response time to activate the main valve of the thermostat and the dependence of the coolant flow direction control on the engine. This study proposes a novel active engine cooling system incorporating a shape memory alloy-based thermostat. After discussing the operating principles, the governing equations of motion were formulated and analyzed using COMSOL Multiphysics and MATLAB. The results show that the proposed method improved the response time required to change the coolant flow direction and led to a coolant temperature difference of 4.90 °C at 90 °C cooling conditions. This result indicates that the proposed system can be applied to existing internal combustion engines to enhance their performance in terms of reduced pollution and fuel consumption.

## 1. Introduction

Over the last few decades, automotive manufacturers and researchers have focused on improving engine efficiency and reducing pollution from exhaust gases [1]. Developing and updating technologies that can increase the engine efficiency while reducing exhaust emissions are both economically and environmentally advantageous [2]. A great deal of research has been conducted to solve the problem of automotive engine emissions [3]. Various studies have also been conducted on the cooling systems of spark ignition and diesel engines [2]. One of the most important methods is to maintain the optimum temperature range of the engine as recommended by the manufacturer. In most cases, the optimum range of internal combustion engines (ICEs) is between 90 °C and 104 °C. Hence, the control temperature prescribed by the manufacturer is 90 °C. Above this temperature, the thermostat valve must change the direction of the coolant flow from the engine to the radiator. When the engine operates below the maximum allowable temperature, the thermostat remains closed, allowing the cooling system to transfer heat to different engine components [4]. In this case, the aim is to not only increase the operating performance of ICEs and reduce their fuel consumption but also to reduce exhaust emissions by improving combustion conditions [5].

Engine overheating can cause premature deflagration of the compressed mixture [6] and incomplete combustion during the power stroke [7]. Other consequences of an engine operating above the prescribed maximum temperature include a blown cylinder head gasket [8], crankshaft bearing wear, a sticking valve in the guider, busted heater hoses and radiators, reduced lubricating oil properties, piston scuffing, and seizure and breakage of overhead cams or camshafts. All of these result in the loss of engine power and the emission of pollutants from fumes during the exhaust stroke [9]. Controlling the engine coolant temperature within a certain range can improve engine efficiency and reduce or prevent incomplete combustion of the mixture in the cylinder. To maintain the desired temperature range in ICEs, active cooling systems with electrical components have been developed, such as cooling fans and coolant pumps, to replace mechanical components [10]. However, such systems use conventional wax and pellet thermostats to control the direction of the coolant flow. An electrically heated wax thermostat has also been developed to replace conventional wax thermostats. The rotational speeds of the cooling fan and coolant pump are independent of the engine crankshaft speed, which reduces the engine load but increases the electric battery power consumption [11]. In recent decades, engine cooling systems based on mechatronic components and control strategies have exhibited improved performance [12]. The engine-control unit controls the power supplied to the cooling fan and coolant pump. With this technology, fully controlled electric thermostats have been developed and integrated into the cooling system [13]. Xinran and Wagner [14] used an intelligently controlled or smart thermostat that reduced the total cooling power consumption by 14% compared with conventional thermostats installed in a computer-controlled cooling system. Kalelis et al. [3] designed an intelligent cooling system with a control system for the coolant pump, cooling fan, and thermostatic valve. Their results showed a 30% reduction in hydrocarbon composite exhaust emissions compared with conventional and controlled active cooling systems. However, such thermostats are complex models with advanced electronic components; hence, they are expensive and difficult to maintain. After a certain period of operation, they no longer serve their purpose and must be replaced.

This paper proposes a new active engine cooling system that uses a new shape memory alloy-based (SMA) thermostat to improve the time response of opening and closing the thermostat. It changes the flow direction of the coolant without delay and continuously operates within the prescribed temperature range of the engine. The SMA-based thermostat is designed with a simplified model of two springs. The main component of the model is a spring made of a binary SMA that activates and controls the operation of the thermostat. The other spring is a preload or bias spring made of stainless steel that keeps the main valve closed by stretching the SMA spring in the martensite phase and releasing it to open the main valve in the austenite phase. The design of the model is relatively simple, and the springs are affordable. Both springs are installed and work in parallel. In this model, the thermostat operates as an electronically controlled actuator, while the time required for the operating response should be minimized. The application of this innovative thermostat model in an active engine cooling system can improve the regulation of the coolant temperature. Based on the constitutive model of the SMA and stainless-steel springs, a curved beam was established, and finite element analysis was conducted to evaluate the design of the main valve of the thermostat. The active cooling system using the proposed SMA-based thermostat was simulated and compared with the existing active cooling system. The time required to change the coolant flow direction, the maintenance of the engine coolant temperature within the prescribed range, and the pressure of the coolant circulating in the cooling system of an ICE were evaluated.

## 2. Materials and Methods

### 2.1. Active Engine Cooling System Concept

Existing ICEs use advanced technologies such as sensors and DC motors in the cooling system. Control and monitoring systems for the cooling components and coolant flowing into the system are easy to install. The temperature sensor detects the temperature of the engine coolant in the system. The control module of the cooling system makes decisions based on the output of the temperature sensor. The control module of the cooling system makes further decisions based on the outputs of the temperature sensor. In the traditional cooling system, rotating components rotate at the same speed as the engine crankshaft and are not dependent on the temperature of the coolant in the system [15]. When a vehicle is on the road, the engine speed is dependent on the road conditions. However, the engine may have a high coolant temperature even at a low speed [16]. The main task of an active engine cooling system is to operate the engine components based on the coolant temperature instead of the crankshaft speed.

The proposed active engine cooling system is shown in Figure 1a. The system uses DC motors to maintain the rotational motion of the coolant pump and cooling fan. The rotational speeds are independent of the engine speed. The independent rotation of the cooling fan and coolant pump improve the engine power density, performance, and fuel economy by reducing the engine load [17]. However, the operation of both systems is controlled by the cooling system controller. A temperature sensor detects the coolant temperature in the engine cooling system and sends these signals to the control unit of the cooling system. The controller analyzes these signals and determines the required speed of the motors for the coolant pump and cooling fan. The controller also manages the operation of the thermostat, including its opening and closing. The mathematical formulation and MATLAB Simulink components for modeling the system have replaced the practical approaches for the system. Therefore, the coolant pump defined in Equation (1) is expressed as the angular velocity of the motor (ϖ), but its control is dependent on the coolant temperature in the system.
(1)$$\frac{d\bar{\omega }}{dt}=\frac{1}{J}\left\{k_{m}i_{a}-\left(b_{i}+L_{R}V_{c}^{2}\right)\bar{\omega }\right\}$$
where *V_c_* is the fluid volume per radian of shaft rotation, *i_a_* is the armature current, *b_i_* is the inlet impeller width, *L_R_* is the pump lump fluid resistance, *k_m_* is the pump motor torque constant, and *J* is the pump-system moment of inertia.

When an ICE operates within the normal coolant temperature range, the bypass valve in the thermostat opens and allows the coolant to flow from the engine block/engine components back to the coolant jackets through the coolant pump to close the engine cooling circuit, as shown in Figure 1b. In this operation phase, the coolant pump rotates at a low speed, while the cooling fan does not rotate because no coolant is flowing into the radiator. A proportional integral derivative (PID) controller is used to control the speed of the electric motors that operate the coolant pump and cooling fan during the analytical simulation in MATLAB Simulink.

The coolant with a temperature above the maximum preset flows from the engine to the radiator through the thermostat. The radiator cools it down at ambient temperature and flows back into the engine block to cool the engine components, as shown in Figure 1c. The radiator uses the heat transfer law expressed in Equation (2). The temperature difference between the inlet and outlet of the radiator is the heat loss of the radiator during cooling [18]. Because the components of the active cooling system operate as a function of the coolant temperature, the heat transferred from the coolant in the radiator to the environment is temperature and time dependent. The volume of air flowing through the radiator fins causes heat loss inside the radiator [19]. This process utilizes Newton’s cooling law, which describes the transfer of heat energy between a surface and moving fluid at different temperatures (convection) [20].
(2)$$\frac{dT_{r}}{dt}=\bar{m_{c}}C_{c}\left(T_{e}-T_{t}\right)-UA_{r}\left(T_{r}-T_{a}\right)$$
where $C_{c}$ is the coolant-specific heat capacity, $\bar{m_{c}}$ is the coolant mass flow rate, $A_{r}$ is the radiator area, *U* is the heat transfer coefficient, $T_{e}$ is the engine coolant temperature, $T_{a}$ is the outside ambient air temperature, and $T_{r}$ is the coolant temperature measured at the radiator outlet hose.

There is no method to control the airflow through the radiator fins without the use of a cooling fan. The DC motor that drives the cooling fan rotates at a speed controlled by the electronic control module of the cooling system, which is dependent on the temperature of the coolant circulating in the system. When the coolant temperature is high, the cooling fan rotates at a high speed to increase the volume of air at the radiator fins. Equating the first law of thermodynamics with Newton’s law of cooling defines the operating principle of the cooling fan in an engine cooling system, as described in Equation (3). Similar to the coolant pump and power supply of the SMA spring, the cooling fan uses a PID controller in the analytical simulation.
(3)$$T_{r}=T_{a}+\left(T_{e}-T_{a}\right)exp\left(\frac{-h_{c}A}{m_{a}C_{a}}\right)$$
where $\bar{m_{a}}$ is the air mass flow rate, $C_{a}$ is the air-specific heat capacity, $h_{c}$ is the heat transfer coefficient, and *A* is the radiant surface area.

### 2.2. Mechanical Design of the SMA-Based Thermostat Model

The proposed thermostat uses two helical coil springs with the same center axis. The SMA material with superelastic properties is used to produce the main component of the model (SMA spring), which differentiates it from existing models. The SMA spring activates the opening of the thermostat based on the prescribed engine operating temperature. It plays the same role in the thermostat as bimetal, wax, and pellet materials. In addition to the SMA spring, the model has a bias spring made of stainless steel. The bias spring is designed to work in the compression mode while the SMA spring works in tension in its martensite phase. The SMA spring regains its original shape during the austenite phase when the force generated by the SMA spring compresses the bias spring to open the thermostat. Both springs use the same design theory but different parametric design relationships. Table 1 lists the major notations used in the design of the coil springs for both the SMA and bias springs.

The parameters include the shear stress ($\tau_{s}$), shear strain (μ), and shear modulus (*G*). These are the material behaviors used to define the spring working conditions [21].
(4)$$\tau_{s\left(max\right)}=\frac{P}{A}+\frac{P_{t}r_{w}}{J}$$
where P, A, $P_{t}$, $r_{w}$, and *J* are the application load, cross-sectional area, torsion force, wire radius, and polar moment, respectively, expressed as follows:(5)$$A=\frac{\pi d_{w}^{4}}{4},\;P_{t}=\frac{PD_{m}}{2},\;r_{w}=\frac{d_{w}}{2}=\frac{D_{m}}{2C},\;J=\frac{\pi d_{w}^{4}}{32}$$
where $D_{m}$, $d_{w}$, and *C* denote the mean coil diameter, wire diameter, and spring index, respectively. After substituting Equation (4) into Equation (5), and considering the shear stress correction factor ($C_{s}$), the shear stress can be expressed as
(6)$$\tau_{s\left(max\right)}=\frac{8PD_{m}}{\pi d_{w}^{3}}\left(\frac{2C+1}{2C}\right)=\frac{8PD_{m}}{\pi d_{w}^{3}}C_{s}$$

In this design, both the direct shear and changes in the coil curvature effects must be considered. However, in spring calculations, these effects must be excluded from the shear stress correction factor ($C_{s}$) [22] that is expressed in Table 2. The discretized elements of the springs must satisfy the beam element characteristics [23]. To maximize the life cycle of the spring during design, the maximum shear stress was used as the theoretical shear stress of the springs in the model [24]. To prevent failure, the theoretical shear stress must always exceed the working stress of the spring. The shear stress on a helical spring was dependent on the shear modulus and shear strain.

The application load (*P*) used in Equations (4)–(6) represents the total force exerted on the piston head of the thermostat valve, which originated from the pressure of the coolant circulating in the system. This load is expressed as
(7)$$P=P_{r}A-P_{r}a=P_{r}\left(A-a\right)$$
where a is the cross-sectional area of the bypass valve piston, *A* is the cross-sectional area of the piston of the main valve, and $P_{r}$ is the pressure of water at the temperature required to open the thermostat, which is expressed as follows:(8)$$P_{r}=6.1121\times exp\left\{\left(18.678-\frac{T_{A}}{234.5}\right)\times \left(\frac{T_{A}}{257.14+T_{A}}\right)\right\}$$
where *T_A_* is the austenite temperature ($A_{f}$) listed in Table 3. The design conditions of the SMA spring include all known design parameters from the material properties and working environment [24]. The austenite shear modulus and shear strain in Table 3 were used to determine the maximum shear stress and simulate the thermostat.

The loads from the thermostat opening temperature in Equation (7) and shear stress in Equation (6) were applied to the piston head of the main valve. This enabled the calculation of the mean diameter of the coil spring ($D_{m}$) and spring wire diameter ($d_{w}$), which are required to select the springs used in the thermostat. The following parameters are required in the design process: the number of coils (number of active coils $n_{a}$ and total number of coils $n_{t}$), lengths (free length $L_{f}$ and solid length $L_{s}$), and pitch distance (*h*), which are listed in Table 1 for both springs. The formulas to determine these parameters based on the known parameters are expressed as follows:(9)$$n_{a}=\frac{G_{a}d_{w}{}^{4}s}{8PD_{m}{}^{3}}$$
(10)$$h_{\left(SMA\right)}=\frac{L_{f}}{n_{a}+1},h_{\left(Bias\right)}=\frac{L_{f}-2d_{w}}{n_{a}}$$
(11)$$L_{s_{\left(SMA\right)}}=d_{w}\left(\left(n_{a}+2\right)+1\right),L_{s_{\left(Bias\right)}}=d_{w}n_{t}$$
where $L_{s}{}_{\left(SMA\right)}$ is the solid length of the SMA spring, $L_{s}{}_{\left(Bias\right)}$ is the solid length of the bias spring, and *S* is the spring deflection; the other variables are listed in Table 1.

The total number of coils of the SMA spring ($n_{t}$) is one more plus the number of active coils $n_{a}$ calculated by Equation (9). Owing to the different working conditions of the spring and spring ends, the total number of coils of the bias spring ($n_{t}$) is two more plus the number of active coils ($n_{a}$). The solid lengths of the SMA and bias springs differ because of their different working conditions. Both the solid length of the SMA spring ($L_{s}{}_{SMA}$), calculated to support the extension load, and that of the bias spring ($L_{s}{}_{Bias}$) are deformed by compression. The upper end of the SMA spring is a hook connected to a cylindrical mounting material, as shown in Figure 2a. A cylindrical mounting with a rod and bias spring assembly is used to perform tensile loading in the martensitic phase. It also helps move the rod together with the main valve piston and bypass valve piston. The hook length and gap are negligible in the design calculations. 

The bottom end of the spring includes a plane and ground end, which fix the spring to the cylindrical part of the thermostat frame. The bias spring generates a force to extend the SMA spring. However, the bias spring operates under compression, with the top and bottom having squared and ground ends. The load generated in the austenite phase of the SMA spring compresses the bias spring, such that it regains its original shape. Thus, the dimensions of the SMA and bias springs listed in Table 4 are based on previous design processes and the results of Equations (4)–(11). To obtain the characteristics of both springs that fit with the proposed model, we use the design parameters listed in Table 1 and the material properties in Table 3. The springs are shown in Figure 2. The proposed thermostat model has the same basic dimensions as the thermostats available in the market. As a result, the current thermostats serve as the basis of the assumptions made during the design process. The SMA spring with the valve rod is assembled through the two holes on the cylindrical bracket or mounting. These holes also reduce the weight and amount of material in the bracket. The cylindrical mounting and hook end in Figure 2a are made of steel and welded together. They support all the working movements of the SMA spring. When heated in the austenite phase, the SMA spring generates the required force P of 17.36 N to compress the bias spring, as shown in Figure 2b. The pitch distance angle and length of the springs change depending on the operating position of the thermostat. The closed and open positions of the main valve determine the position of the thermostat.

### 2.3. Finite Element Formulation

The global coordinate system (*X*, *Y*, and *Z*) defines a single coil of the helical spring, as shown in Figure 3a. The perpendicularity of the succession plane domains to the curvilinear length (*dx*) provides the global coordinate system. This helical spring has a constant inner coil radius (*R*) and a uniformly distributed load (*P*_0_) on its curved helical beam. The position vector $\vec{R}$ with angular coordinate ψ in Equation (12) describes a point on the curved coil spring element. The curvilinear length (*dx*) and orthonormal unit vectors (*u, v, w*) in the direction of the global coordinate axis are given by Equations (13) and (14). The orthonormal unit vectors (*u*, *v*, and *w*) represent the unit tangent vector to the curve, the unit normal vector to the curve, and the binormal vector, respectively.
(12)$$\vec{R}=R\left(\cos\psi \vec{i}+\sin\psi \vec{j}+\frac{h}{2\pi }\psi \vec{t}\right)$$
(13)$$dx=\left|\frac{d\vec{R}}{d\psi }\right|=\Gamma d\psi$$
where
$$\Gamma =\sqrt{R^{2}+{\left(\frac{h}{2\pi }\right)}^{2}}$$
(14)$$\begin{array}{l} u=\frac{d\vec{R}}{dx}=\left(-R\sin\psi \vec{i}+R\cos\psi \vec{j}+\frac{h}{2\pi }\vec{t}\right)\Gamma^{-1}\\ v=\frac{d\vec{R}}{dy}=-\cos\psi \vec{i}-\sin\psi \vec{j}\\ w=u\times v=\left(\frac{h}{2\pi }\sin\psi \vec{i}-\frac{h}{2\pi }\cos\psi \vec{j}+R\vec{t}\right)\Gamma^{-1} \end{array}$$
(15)$$\begin{array}{l} R_{c}={\left\|\frac{d{\vec{e}}_{1}}{dx}\right\|}^{-1}=\frac{\Gamma^{2}}{R}\\ R_{t}={\left\|\frac{d{\vec{e}}_{3}}{dx}\right\|}^{-1}=\Gamma^{2}\frac{2\pi }{h} \end{array}$$
where the curvature radius $R_{c}$ is reciprocal to the curvature of the helical beam, *h* is the spring pitch distance between the two coils; *u*, *v*, and *w* are the displacement components in the *X*-, *Y*-, and *Z*-directions, respectively; and $R_{t}$ is the torsional radius that is reciprocal to the torsional radius of the space curve of the coil, as shown in Figure 3a. The moderated analysis of the model uses numerical techniques and mathematical formulations [25], which significantly differ from the analysis of the deformation behavior of the two springs. The stainless-steel spring exhibits normal deformation behavior [26] while the SMA spring exhibits normal deformation and superelastic shear behaviors [27]. The characteristic behaviors of SMA materials are independent of one another [28]. Figure 3b shows the nonlinear Euler–Bernoulli beam describing the spring element. It defines, in detail, the finite element formulation and finite element analysis of springs used in this model. The displacement fields of both the linear and nonlinear parts of the beam and constitutive equations will be required to reach the entire spring element hysteresis for both stress vs. strain and force vs. displacement.

The displacement magnitudes of the pair of springs must reflect the total stroke at the full opening and closing of the main valve. A stroke is defined as the difference between the height of the bias spring under the initial condition (*L_f_*) and its solid length (*L_s_*). From Equations (9) and (11), the maximum stroke at the full opening of the thermostat is 7.8 mm. However, the finite element formulation for large displacements was adapted for use in this model. The displacement of the entire spring refers to the stroke at the full opening and closing of the main valve. For the analytical description of the spring, the virtual work principle in Equation (16) can express the equation of motion for the Euler–Bernoulli beam to identify the working characteristics of the model.
(16)$$\int_{V}\sigma_{xx}\delta \varepsilon_{xx}^{T}dv=\int_{S}P_{0}\delta wds$$
where σ ε are the normal stress and Green–Lagrange strain tensor for undeformed coordinates along the horizontal centroid axis, respectively, $P_{0}$ is the tension load, and w is the displacement. 

The normal stress is a product of the strain (ε) and Young’s modulus (*E*) that defines the material behavior under analysis [29]. However, this principle only works for the formulation of the stress–strain relationship in a finite element for stainless steel and other materials without superelastic properties, such as SMA materials. The stresses of the SMA and stainless-steel beam elements are different. The strain energy cannot be reached without founding an axial load. Therefore, the normal stress and strain tensors in Equation (16) are replaced with their values. The strain tensor includes the field displacement in Figure 3b and considers the left-hand term of the virtual work principle. We obtain this in a new form with two terms in the function of the axial load and bending moment, as expressed in Equation (17).
(17)$$\int_{V}E\varepsilon_{xx}\delta \varepsilon_{xx}dv=\int_{0}^{L}\int_{A}E\varepsilon_{xx}\delta \varepsilon_{xx}dAdx=P\int_{0}^{L}\left(\delta \frac{\partial u_{0}}{\partial x}+\frac{\partial w}{\partial x}\delta \frac{\partial w}{\partial x}\right)dx+M\int_{0}^{L}\delta \frac{\partial^{2}w}{\partial x^{2}}dx$$
where P is the axial load, $u_{0}$ is the axial displacement, w is the transversal displacement, A is the cross-sectional area, M is the bending moment, and L is the beam length. Both *P* and *M* are expressed in detail in Equation (18). These were used to determine the nodal displacement and stiffness matrices of the beam.
(18)$$\begin{array}{l} P=\int_{A}\sigma dA=\int_{A}E\left(\frac{\partial u_{0}}{\partial x}-z\frac{\partial^{2}w}{\partial x^{2}}+\frac{1}{2}{\left(\frac{\partial w}{\partial x}\right)}^{2}\right)dA=EA\left(\frac{\partial u_{0}}{\partial x}+\frac{1}{2}{\left(\frac{\partial w}{\partial x}\right)}^{2}\right)\\ M=-\int_{A}\sigma zdA=\int_{A}Ez\left(\frac{\partial u_{0}}{\partial x}-z\frac{\partial^{2}w}{\partial x^{2}}+\frac{1}{2}{\left(\frac{\partial w}{\partial x}\right)}^{2}\right)dA=EI\frac{\partial^{2}w}{\partial x^{2}}\\ \left[\begin{array}{l} P\\ M \end{array}\right]=\left[\begin{array}{l} EA\\ 0 \end{array}\right.\left.\begin{array}{l} 0\\ EI \end{array}\right]\left[\begin{array}{l} \frac{\partial u_{0}}{\partial x}+\frac{1}{2}{\left(\frac{\partial w}{\partial x}\right)}^{2}\\ \frac{\partial^{2}w}{\partial x^{2}} \end{array}\right]=[K_{e}][D_{L}+D_{NL}]w_{e} \end{array}$$
where *A* is the cross-section of the beam; *I* is the moment of inertia of the beam defined by the h and b of the beam; *K_e_* is the element stiffness matrix; *D_L_* and *D_NL_* are the strain nodal displacement matrices of the beam element for both the linear and nonlinear parts, respectively; and *W_e_* is the nodal displacement vector. 

The two springs in the model are analyzed using the same procedure but with different boundary conditions and working characteristic parameters. As in the in-plane stress analysis, the normal and shear deformations are similar but with different mathematical formulations [30]. The expansion of the shear stress from Equation (5) provides the basic equation for the numerical analysis of the SMA spring in the model. Hence, the finite element analysis of this model deals more deeply with the SMA spring compared with the other parts of the thermostat. Unlike the stainless-steel spring, the SMA spring element must include both the normal and shear stress in the analysis. The shear stress $\tau_{s}$ and normal stress σ of the SMA beam element are expressed by Equations (19) and (20), respectively.
(19)$$\tau_{s}=\tau_{s_{0}}+G_{s}\left(\left(\mu -\mu_{0}\right)-\mu_{R}\left(\upsilon_{s}-\upsilon_{s_{0}}\right)\right)$$
(20)$$\sigma =\sigma_{0}+E\left(\left(\varepsilon -\varepsilon_{0}\right)-\varepsilon_{R}\left(\upsilon -\upsilon_{0}\right)\right)+\beta \left(T-T_{0}\right)$$
where
$$G_{s}=\left(\frac{\upsilon_{s_{*}}}{G_{a}}G_{m}-\upsilon_{s_{*}}+1\right)G_{a}\;\mathrm{and}\;E=\left(\frac{\upsilon_{{}_{*}}}{E_{a}}E_{m}-\upsilon_{{}_{*}}+1\right)E_{a}$$
Note that $\upsilon_{s}$ is the shear stress induced by the martensitic volume fraction, υ is the martensitic volume fraction due to normal stress, *T* is the coolant temperature, β is the thermal elastic coefficient, and (**_0_**) denotes the initial and other parameters defined in Table 3.

The beam layers in Figure 3c describe the shapes of the elements after meshing, which indicate the fine or coarse discretization of the beam elements. The element shape and number of elements in beam meshing play an important role in the output results. The rectangular discretized element shown in Figure 3d has two nodes (1 and 2), as described by the shape function for every node. The shape functions of the beam element through the natural coordinate system (η) define the axial displacement and slope of each node, both of which influence the stress–strain and force–displacement relationships of the model. The differential of the total shear stress–strain and total normal stress–strain can provide the incremental stress–strain relationship used in analytical studies of SMA springs [31]. The stress-induced martensitic volume fraction of the spring and the parameterization characteristics in Table 3 define the phase transformation of the SMA materials in the mathematical formulations of the nonlinear beam element and entire SMA spring. 

Both the martensitic and austenite phases are considered in the SMA spring analysis to determine the performance of the spring in the thermostat. The strain energy and external load applied to the beam indicate its motion characteristics, which were determined by Equations (18)–(20). The designed model could operate under fixed boundary conditions. The malfunction of the system could be attributed to the change in the operation of the springs. The first working condition is the closed main valve when the coolant temperature remains below 90 °C. The second is the opening of the main valve each time the coolant temperature is equal to or greater than 90 °C. The shear stress (τ_s_) and shear strain (*µ*) of the SMA coil spring contribute to its superelastic behavior, along with the torsional deformation. The mechanical characteristics of the SMA coil spring are defined by the normal stress (σ) and normal strain (ε) in combination with the axial and bending deformation [32].
(21)$$\begin{array}{l} d\tau_{s}=\frac{dG_{s}}{d\upsilon_{s_{*}}}\left(\frac{\partial \upsilon_{s_{*}}}{\partial \tau_{s}}d\tau_{s}+\frac{\partial \upsilon_{s_{*}}}{\partial T}dT\right)\left(\mu -\mu_{0}\right)+G_{s}d\mu \\ -\left(\frac{G_{s}d\mu_{R}}{dG_{s}}\frac{dG_{s}}{d\upsilon_{s_{*}}}\left(\frac{\partial \upsilon_{s_{*}}}{\partial \tau_{s}}d\tau_{s}+\frac{\partial \upsilon_{s_{*}}}{\partial T}dT\right)\left(\upsilon_{s}-\upsilon_{s_{0}}\right)+G_{s}\mu_{R}\left(\frac{\partial \upsilon_{s}}{\partial \tau_{s}}d\tau_{s}+\frac{\partial \upsilon_{s}}{\partial T}dT\right)\right) \end{array}$$
(22)$$\begin{array}{l} d\sigma =\frac{dE}{d\upsilon_{*}}\left(\frac{\partial \upsilon_{*}}{\partial \sigma }d\sigma +\frac{\partial \upsilon_{*}}{\partial T}dT\right)\left(\varepsilon -\varepsilon_{0}\right)+Ed\varepsilon +\beta dT\\ -\left(\frac{Ed\varepsilon_{R}}{dE}\frac{dE}{d\upsilon_{*}}\left(\frac{\partial \upsilon_{*}}{\partial \sigma }d\sigma +\frac{\partial \upsilon_{*}}{\partial T}dT\right)\left(\upsilon -\upsilon_{0}\right)+E\varepsilon_{R}\left(\frac{\partial \upsilon }{\partial \sigma }d\sigma +\frac{\partial \upsilon }{\partial T}dT\right)\right) \end{array}$$

The derivation of terms in Equations (22) and (23) by considering the temperature of both the martensitic and austenite phases leads to the incremental stress–strain relations of the SMA beam element.
(23)$$\begin{array}{l} By\_\tau_{{}_{s}}\Rightarrow G_{s}d\mu +\left(\frac{dG_{s}}{d\upsilon_{s_{*}}}\frac{\partial \upsilon_{s_{*}}}{\partial T}\left(\mu -\mu_{0}\right)-\left(\frac{G_{s}d\mu_{R}}{dG_{s}}\frac{dG_{s}}{d\upsilon_{s_{*}}}\frac{\partial \upsilon_{s_{*}}}{\partial T}\left(\upsilon_{s}-\upsilon_{s_{0}}\right)+G_{s}\mu_{R}\frac{\partial \upsilon_{s}}{\partial T}\right)\right)dT\\ By\_\sigma \Rightarrow Ed\varepsilon +\left(\frac{dE}{d\upsilon_{*}}\frac{\partial \upsilon_{*}}{\partial T}\left(\varepsilon -\varepsilon_{0}\right)-\left(\frac{Ed\varepsilon_{R}}{dE}\frac{dE}{d\upsilon_{*}}\frac{\partial \upsilon_{*}}{\partial T}\left(\upsilon -\upsilon_{0}\right)+E\varepsilon_{R}\frac{\partial \upsilon_{*}}{\partial T}\right)+\beta \right)dT \end{array}$$
where $\varsigma_{s}^{cr}\;\mathrm{and}\;\varsigma_{f}^{cr}$ are the critical starting and finishing stress of martensite transformation; *As* and *A_f_* are the austenite starting and finishing temperature; *M_s_* and *M_f_* are the starting and finishing martensitic temperature; $\phi_{m}\;\mathrm{and}\;\phi_{a}$ are the slope induced between the critical transformation stress and temperature, respectively; and $\sigma_{r}$ is the von Mises equivalent stress. The analysis of the incremental stress–strain relationship of a helical spring made from an SMA material must use Equation (24).
(24)$$\left\{\Delta \sigma \right\}=\left[\kappa_{se}\right]\left(\left\{\Delta \xi \right\}-\left\{\Delta \xi_{se}\right\}\right)$$
where
$$\left\{\Delta \sigma \right\}=\left\{\begin{array}{l} \Delta \sigma \\ \Delta \tau_{s_{xz}}\\ \Delta \tau_{s_{yz}}\\ \Delta \tau_{s} \end{array}\right\},\left\{\Delta \xi \right\}=\left\{\begin{array}{l} \Delta \varepsilon \\ \Delta \mu_{xz}\\ \Delta \mu_{yz}\\ \Delta \mu \end{array}\right\},\left\{\Delta \xi_{se}\right\}=\left\{\begin{array}{l} \Delta \varepsilon_{se}\\ 0\\ 0\\ \Delta \mu_{se} \end{array}\right\},\left[\kappa_{se}\right]=\left[\begin{array}{cccc} E_{se} & 0 & 0 & 0\\ 0 & G_{s} & 0 & 0\\ 0 & 0 & G_{s} & 0\\ 0 & 0 & 0 & G_{s_{se}} \end{array}\right]$$

The load–displacement relationship is used to analyze the designed system based on SMA materials [33]. Based on the virtual work principle, the internal and external loads can be determined by solving the internal and external work, respectively. Virtual work is different from real work, although they are similar. Equating the external and internal work or strain energy expressed in Equation (26) defines the principle of real work and helps determine the external load used in the load–displacement curve of the SMA spring. If the external load is equal to the internal load, then the residual force is zero.
(25)$$\begin{array}{l} U_{ext}=\frac{1}{2}\sum_{i=1}^{n}P_{i}w_{i}=\frac{1}{2}{\left\{P\right\}}^{T}\left\{w\right\}\\ U_{\mathrm{int}}={\int_{V}\left\{\sigma_{xx}\right\}}^{T}\left\{\varepsilon_{xx}\right\}dV \end{array}$$
where $U_{ext}$ and $U_{int}$ are the external work carried out by force P and the internal work or total strain energy carried out by P, respectively.

The simulation of the spring in line with Equations (18), (25), and (26) by a combination of the different acting amplitudes of the spring coils yielded a cyclic load–displacement relation in the martensite phase. However, the austenite phase of the SMA spring deactivated the mechanical behavior of the martensite phase, which caused the spring operation to be dependent on the prescribed final temperature used to heat it [34]. Although it responds to the temperature of the engine coolant, this temperature cannot activate it because the final austenite temperature ($A_{f}$) is 94 °C. The prescribed maximum operating temperature of the engine coolant is 90 °C, which was selected during the simulation. The electric current source heated the SMA spring to 94 °C. The electric current (*I*) required to heat the SMA spring at the final austenite temperature was calculated by Equation (26) to generate the force necessary to compress the bias spring or extend it to its full length ($L_{s}$).
(26)$$H=I^{2}Rt$$
where H, R, and t are the heat capacity of the SMA spring, austenite electrical resistivity, and heating time, respectively.

### 2.4. SMA-Based Thermostat Operating Mechanism

Figure 4a shows the basic components of the SMA-based thermostat. It contains two valves: the main valve and bypass valve, which operate in opposite directions. The bypass valve is dependent on the operation of the main valve [35]. A rod mounted on the thermostat bridge links the two valves through the frame base. The main valve piston is in contact with the valve seat, which has a rubber sealing material to prevent the coolant or pressure from leaking into the system when the thermostat is closed. The bypass valve closes when the thermostat is open. Because the thermostat is constantly immersed in water [36], its components are made of stainless steel to prevent rust [37], except for the SMA spring. The SMA spring has a particular task in the thermostat to activate the control of the coolant flow direction based on the prescribed range of the engine operating temperature within a defined time. The response time of SMA actuators is in the range of 0.1 to 6 s [38]. The thermostat and other components of the engine cooling system ensure that the engine does not overheat and maintains an efficient operating temperature [39]. In an ICE, the normal operating temperature of the coolant is between ambient temperature (30 °C) and 100 °C. The engine can operate above this temperature but not for long to avoid overheating [40]. In this study, the maximum prescribed temperature is 90 °C. The two-way SMA spring installed in the main valve plays an important role in controlling the opening and closing of the thermostat to allow the coolant to flow from the engine to the radiator or circulate further into the engine components. In the martensite phase of the SMA spring, the coolant temperature is between 20 °C (M_S_) and 82 °C (M_f_). In this phase, the bias spring is in its original or ideal position. The rod links the main and bypass valves through the cylindrical mounting part of the hook end of the SMA spring assembly. The SMA and bias springs work in opposite directions. The SMA spring extends to move the full stroke (7.8 mm) and allows the bias spring to hold the main valve piston head fully in the valve seat. This closes the main valve and opens the bypass valve, as shown in Figure 4b.

The coolant flows back into the engine block (coolant jackets) through the bypass valve, as indicated by the blue arrows. When the coolant temperature reaches the specified maximum normal coolant operating temperature (90 °C), an electric current is applied to heat the SMA spring to its activation temperature (94 °C, *A_f_*), as shown in Figure 5. The SMA spring overcomes the tension of the bias spring to regain its original shape and position. It compresses the bias spring to release the main valve piston, opens the main valve, and closes the bypass valve, as shown in Figure 4c. The coolant flows from the engine through the main valve of the thermostat to the radiator, as indicated by the red arrows. The electric current supplied to the SMA spring is stopped after its activation. The coolant in the system begins to cool back to the martensite phase. This reduces the power consumption, prevents the SMA spring from overheating, and ensures that the SMA spring performs well for multiple operating cycles [41]. The SMA-based thermostat allows the activation temperature to be set at variable ranges according to the engine manufacturer’s requirements.

## 3. Results and Discussions

### 3.1. Thermostat Closing and Opening Characterization

Figure 6a shows a complete SMA-based thermostat with the position of the main valve and the bypass valve. The main valve is closed, and the bypass valve is opened. This means that the engine is operating at its normal operating temperature. The valve positions should change in a reverse way when the prescribed maximum temperature is noticed. However, the cross-sectional area of the piston head for the main valve is different from the one for the bypass valve. The difference in size results in a pressure difference inside the thermostat within both positions. In the closed position, the pressure pulls the piston head of the main valve out of the valve seat. The bias spring in the idle position ensures that the piston head remains in contact with the valve seat. This occurs after the initial opening and closing of the thermostat. The engine cannot reach its normal operating temperature because of the low pressure of the coolant in the cooling system. The finite element analysis results from COMSOL Multiphysics, shown in Figure 6, describe the effects of such depressurization in the main valve of the thermostat. Both the main and bypass valves have the same direction but different operating positions. The concave shape of the valve seat and piston head shape allow the piston head to have a small downward displacement, as shown in Figure 6b. The maximum deflection or displacement that occurred at the piston head crown was $9.76\times 10^{-8}$ mm. However, this amount of deflection may not cause the operating failure of the main valve. The valve head and seat remained in contact with each other. The lack of contact between the valve seat and piston head may cause the thermostat and engine to malfunction. The malfunction of the thermostat leads to insufficient pressure and low engine performance. The maximum stress of $1.5\times 10^{3}$ N/m^2^ shown in Figure 6c was induced on the main valve when it was closed. This stress occurred after the allowable design stress of the SMA spring in the martensite phase. It is also lower than the static yield strength of the thermostat material obtained by static analysis ($2.827\times 10^{8}$ N/m^2^). The part of the cylindrical mounting in contact with the rod was highly stressed. The maximum strain due to the piston deflection shown in Figure 6d was $4.55\times 10^{-9}$ at the piston head near the rod. The deflection and elongation or strain also played important roles in the operating mechanism of the thermostat. They prevented the piston head from sticking to the valve seat and accelerated the opening of the thermostat after the activation of the SMA spring. The complete closing of the main valve led to the full opening of the bypass valve and promoted the circulation of the coolant in the system.

In the martensite phase, the load generated by the bias spring at its optimum position elongated the SMA spring. The SMA spring moved across the entire free length of the bias spring and then returned to its initial position when the applied load was removed. The force-displacement curve returned to the origin while retaining hysteresis, as shown in Figure 7a. The superposition in the force-displacement curve resulted from the number of coils of the bias spring and the amplitude of the force acting on the SMA spring. The SMA spring was studied separately and moved five times more than the total stroke when installed in the thermostat valve. The stress–strain curve in Figure 7b shows the superelastic characteristics of the SMA spring. Its maximum strain in the martensite phase was $5\times 10^{-4}$ when extended to reach the full stroke. The small hysteresis of the SMA spring under loading conditions is important for the design process. It indicates the high operating capacity of the selected material properties, which allowed the spring to operate continuously for several cycles without failure. The thermostat was opened to change the direction of the coolant flow, which then flowed from the engine to the radiator. When the prescribed maximum temperature of the engine was reached, the coolant flowed into the radiator. The radiator cooled the coolant to the ambient temperature, which then flowed back into the engine through the coolant pump. At this position of the thermostat, the SMA spring operated in the austenite phase. The coolant temperature and electric current were required to heat the steel to its final austenite temperature. This created a force that compressed the bias spring to its entire solid length, while the SMA spring returned to its initial state, as shown in Figure 8. Various forces and strokes were observed between the first and last cycles of the SMA spring. The generated force corresponded to the electric current supplied to the SMA spring.

As listed in Table 4, the force required to compress the bias spring to its solid length until the thermostat is fully open was 17.36 N. However, the SMA spring only produced a maximum force of 16 N on the first cycle and 17.25 N on the last cycle. The force difference of 0.11 N ensured the safe operation of the piston head because of pressure variation after the opening process of the main valve. The electrical current input was 0.7 A, which is the highest current used in the simulation. This allowed the piston head of the main valve to perform strokes of 7 mm and 7.8 mm, as shown in Figure 8. The supply of 0.5 A to the SMA spring generated a maximum force of 12 N to compress the bias spring and moved the piston of the main valve by a maximum of 5.9 mm. The lowest force produced by 0.3 A of electric current was 2.05 N, and the stroke was between zero and $1.07\times 10^{-34}$ mm. The curve of the smallest stroke in this figure nearly coincides with the horizontal axis. The generated force and stroke of the spring in the last cycle improved compared with those in the other cycles, owing to the heat stored in the SMA spring. In the final cycles, i.e., between the 10th and 20th, the maximum stroke and force were achieved. The maximum stroke led to the complete opening of the main valve and the complete closing of the bypass valve.

### 3.2. Active Engine Cooling System with New Model Thermostat

The operating heat of an ICE varies depending on the stroke of the engine’s operating cycle, as shown in Figure 9. From the compression stroke to the power stroke, the engine heat continuously increased. The greatest amount of heat was generated in the combustion chamber during the power or combustion stroke. The heat decreased in the intake stroke when fresh air and fuel entered the cylinder, and in the exhaust stroke when the burnt mixture was expelled from the engine to the environment. The engine temperature increased as the engine heat increased. The temperature in the combustion chamber after the power stroke could reach 2095 °C and 3315 °C for diesel and spark ignition engines, respectively. The longer an engine runs, the more heat is generated. During the simulation, the highest heat measured was $3.5\times 10^{4}\;$ J/K at $11\times 10^{3}$ S, as shown in Figure 9. When the engine operated within its normal temperature range, the engine cooling system transferred heat from the combustion chamber to the other parts of the engine. The operating temperature of the engine was also maintained, which optimized its performance. The engine cooling system removed excess heat from the engine and released it into the atmosphere. If the thermostat opening is delayed, the engine could overheat; if it opens too soon, the engine could run below the normal operating temperature.

Figure 10 shows the timing response of the mechanical wax, electrically heated wax/solenoid, and SMA-based thermostats. The thermostats responded to the engine temperature to keep the system operating for the designated purposes. However, the SMA-based thermostat responded at 0.26 s to open the main valve of the thermostat. The coolant flowed through the upper radiator hose from the engine to the upper tank of the radiator. The other types of thermostats could have a delayed response to changes in the direction of the coolant flow. They could open the main valve between 6 and 30 s after the prescribed opening temperature is reached. The performance of the solenoid/electric thermostat was better than that of the mechanical wax thermostat, as shown in Figure 10. The coolant temperature above 90 °C must flow from the engine block to the radiator, but this could happen when the thermostat opens. It mixes with the cold water in the upper tank of the radiator. The heat distribution of the coolant in the radiator, shown in Figure 11, describes the cooling phenomena and heat transfer in the radiator. The coolant in the upper tank of the radiator (in yellow) was hotter than that (in blue) in the lower tank of the radiator. The difference in the density and temperature of the coolant caused it to separate into different directions. The coolant with a high temperature (50–85 °C) had a low density, while that with a low temperature (below 50 °C) had a higher density. Therefore, the cold water/coolant can flow from the lower tank of the radiator through the coolant pump into the engine block to cool the engine. The higher the temperature of the coolant in the radiator, the higher the fan speed required to increase the volume of air exiting through the radiator fins. This distinguishes the actively controlled fan from a mechanical one.

### 3.3. Operations of the Current and Newly Proposed Active Engine Cooling Systems 

We compared the existing engine cooling system and the proposed active engine cooling system in terms of their operating temperature and pressure. The operating temperature curves of the engine with two active cooling systems are shown in Figure 12. During the simulation, the prescribed operating temperature range of the engine was 0–90 °C. When the engine temperature was between 0 and 90 °C, the thermostat kept the main valve closed and opened the bypass valve (closed thermostat) until the engine coolant temperature was above the prescribed maximum temperature (90 °C). This maintained the normal operating temperature of the engine. However, above the prescribed maximum temperature (90 °C), the main valve must be opened by the thermostat while the bypass valve must be closed (open thermostat). The coolant flowed from the engine to the radiator to prevent the engine from overheating. Because delays could occur when the thermostat opens, there is some tolerance for temperatures above the prescribed maximum temperature. The active engine cooling system, which uses a wax thermostat, had a delay in changing the direction of the coolant flow. When the opening of the thermostat was delayed, the coolant continued to circulate through the bypass valve in the engine. The coolant temperature slightly increased to 100 °C with this system. When the valve opened, the temperature of the coolant in the system was regulated to 94.90 °C, as shown by the red curve in Figure 11. The time required to activate the opening of the SMA-based thermostat was shorter than that required for the wax thermostat, as shown in Figure 10. Therefore, the temperature did not exceed the tolerance range. The active cooling system with the SMA-based thermostat regulated the coolant temperature to maintain it within the prescribed range, as shown by the blue curve in Figure 12. The thermostat was kept open to prevent the coolant from flowing back into the engine at the prescribed maximum temperature. The difference in the operating temperature between the two active engine cooling systems was 4.90 °C. The coolant temperature was maintained within the normal operating range using the active engine cooling system with the new thermostat model. Within the prescribed coolant temperature range, the engine with the new active cooling system did not operate under overheated conditions.

The heat absorbed by the engine throughout its operation cycle was released to various engine components and even into the atmosphere, causing pressure changes in the cooling system. The higher the temperature of the coolant circulating in the system, the higher its pressure. Figure 13 shows the pressure of the cooling system. Before the coolant temperature reached the thermostat’s opening temperature, the coolant pressure was the same in both systems. The pressure differences occurred after the opening of the thermostat. The active engine cooling system with an SMA-based thermostat opened the thermostat without delay. When the thermostat was opened, the pressure decreased depending on the coolant flow rate and coolant temperature in the system. The SMA-based thermostat maintained the specified temperature range in the engine cooling system, which also maintained the normal variations in the coolant pressure. In contrast, the active cooling system, which used a wax thermostat, delayed the opening of the main valve, which caused the coolant temperature to continuously rise and led to fluctuations in the pressure of the system, as shown in Figure 13. The simulation results showed that the pressure difference between the two active systems was approximately 684 Pa. However, this was the average pressure measured in the systems of up to $12\times 10^{3}\;$ S, where the pressure was highest in both systems. Both the temperature and pressure of the engine coolant are a function of time.

## 4. Conclusions

This study simulated an active engine cooling system using a novel thermostat valve. The aim of this innovative cooling system is to quickly change the direction of the coolant flow and maintain the coolant temperature within the prescribed temperature range. The thermostat response time was shorter than those of other active engine cooling systems. The system adapted to the specified operating temperature of the engine and could be installed in any type of ICE. The results of the simulations show that the SMA-based thermostat can open the main valve in a time range of 0.26 s. The active cooling system with this thermostat has uniform temperature control. The system regulates the temperature so that the thermostat opens whenever the temperature rises above 90 °C. In the simulation, the prescribed maximum temperature was 90 °C. The measured operating temperature difference between the active engine cooling system with the wax thermostat and that with the SMA-based thermostat was 4.90 °C under the same engine operation conditions. The operating coolant pressure difference between the two cooling systems was 697$\;N/m^{2}$. The wax thermostat took 29.06 s longer than the SMA-based thermostat to fully open the main valve. With 0.7 A of electric current supply, the maximum force generated by the SMA spring was 17.25 N, and the main valve piston head moved 7.8 mm of displacement.

The engine cooling system proposed in this work has several advantages, such as stable pressure in the upper radiator hose, the engine operating at a consistent temperature, the required electric current being less than 1 A to activate the thermostat valve, and the engine components not wearing from engine overheating. Therefore, to summarize, the operating engine efficiency is enhanced, fuel consumption is low, and exhaust emissions are reduced by utilizing the proposed system. Finally, future work will include experiments and the practical implementation of this method in a real internal combustion engine to validate the simulation results and identify the potential improvements to the engine cooling system with new technology.

## Figures and Tables

**Figure 1 sensors-23-03972-f001:**
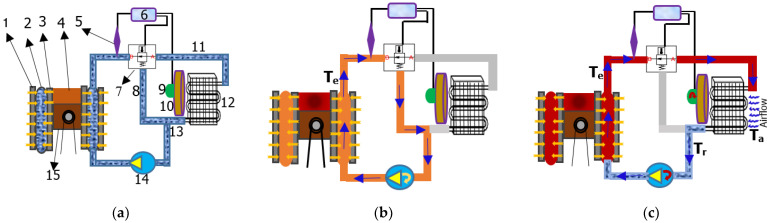
Active engine cooling system. (**a**) Simplified system architecture: (1) engine block, (2) cooling jacket, (3) cylinder wall, (4) combustion chamber, (5) temperature and pressure sensor, (6) cooling system control module, (7) SMA-based thermostat, (8) bypass hose, (9) electrical motor, (10) fan, (11) radiator upper hose, (12) radiator, (13) radiator lower hose, (14) coolant pump, and (15) piston. (**b**) System with closed thermostat (the blue arrows indicate the flow direction of the coolant in the system.). (**c**) System with open thermostat (*Ta*: outside ambient air temperature, *Te*: engine coolant temperature, *Tr*: coolant temperature measure at radiator outlet hose).

**Figure 2 sensors-23-03972-f002:**
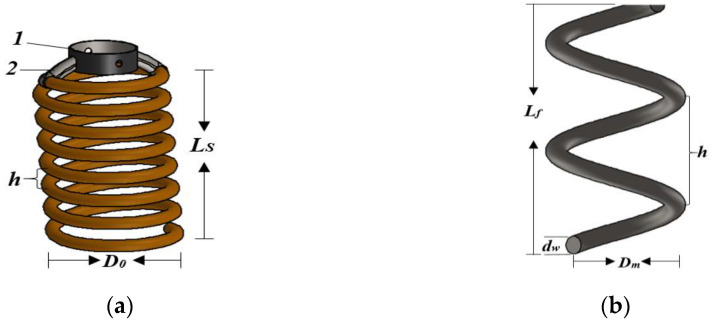
Structural schematics of springs: (**a**) SMA spring with cylindrical mounting 1 on the hook end 2; and (**b**) bias spring.

**Figure 3 sensors-23-03972-f003:**
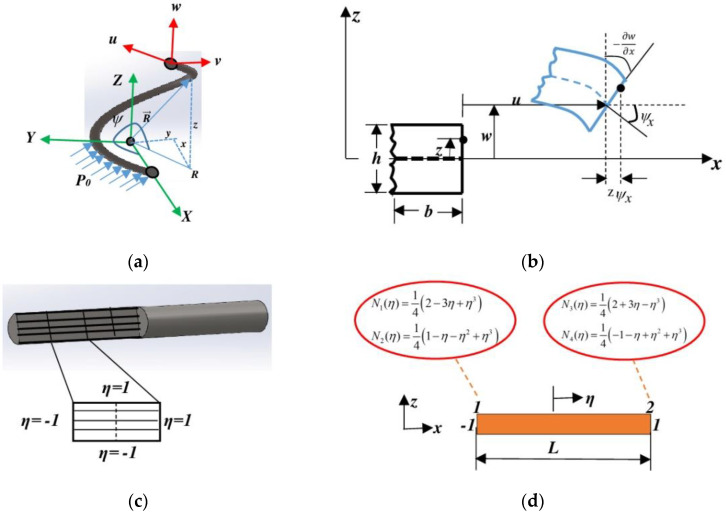
Finite element model: (**a**) helical curve under radial load in a local coordinate system, (**b**) kinematics of nonlinear Euler–Bernoulli beam, (**c**) layered beam element, and (**d**) beam element in the natural coordinate system.

**Figure 4 sensors-23-03972-f004:**
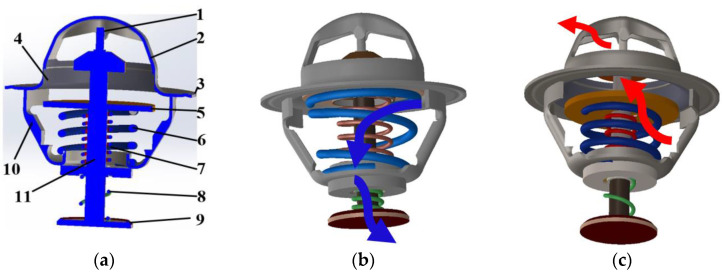
Schematic of SMA-based thermostat. (**a**) Components of the SMA-based thermostat: (1) thrust pin, (2) bridge, (3) flange, (4) valve seat, (5) main valve piston, (6) bias spring, (7) SMA spring, (8) bypass spring, (9) bypass valve piston, (10) frame, and (11) rod. (**b**) Closed thermostat (the blue arrows represent the coolant flow direction when the thermostat is closed). (**c**) Open thermostat (the red arrows indicate coolant flow once the thermostat is opened).

**Figure 5 sensors-23-03972-f005:**
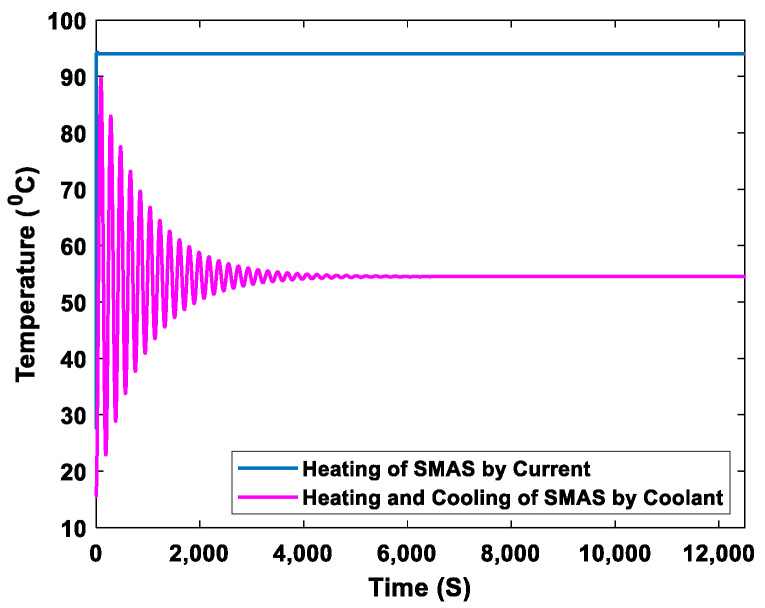
Heating process of the SMA spring.

**Figure 6 sensors-23-03972-f006:**
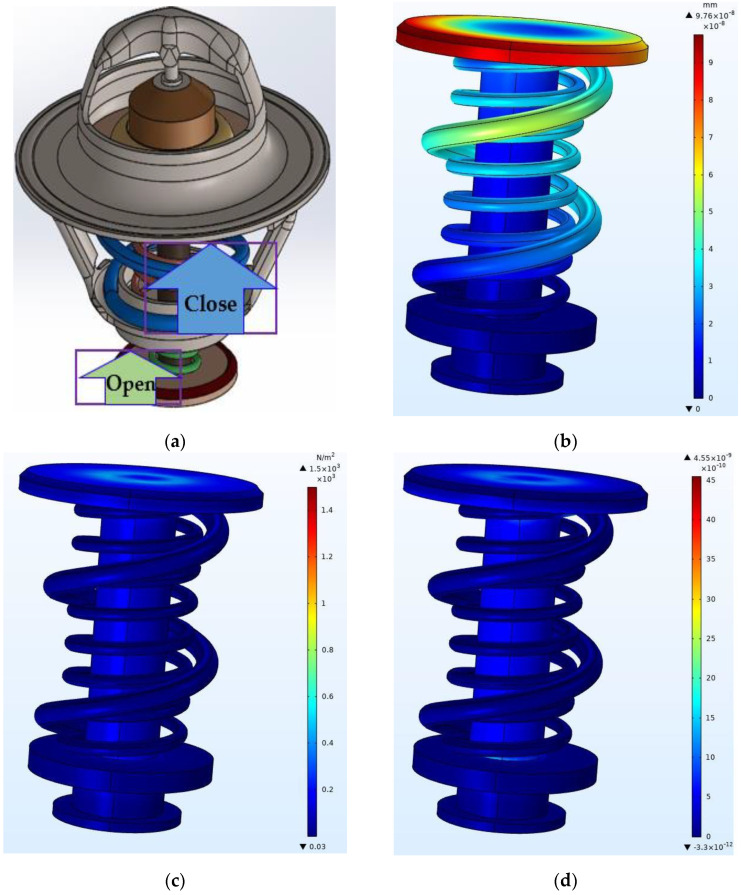
Static finite element analysis of the main valve. (**a**) Thermostat in a closed position, (**b**) displacements, (**c**) von Mises stresses, and (**d**) strain finite element model.

**Figure 7 sensors-23-03972-f007:**
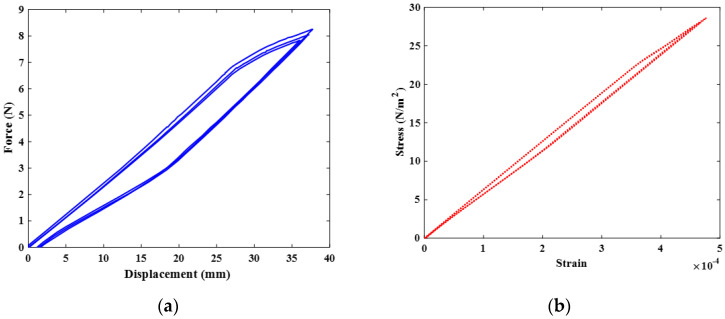
Hysteresis of SMA: (**a**) force-displacement curve and (**b**) stress–strain curve.

**Figure 8 sensors-23-03972-f008:**
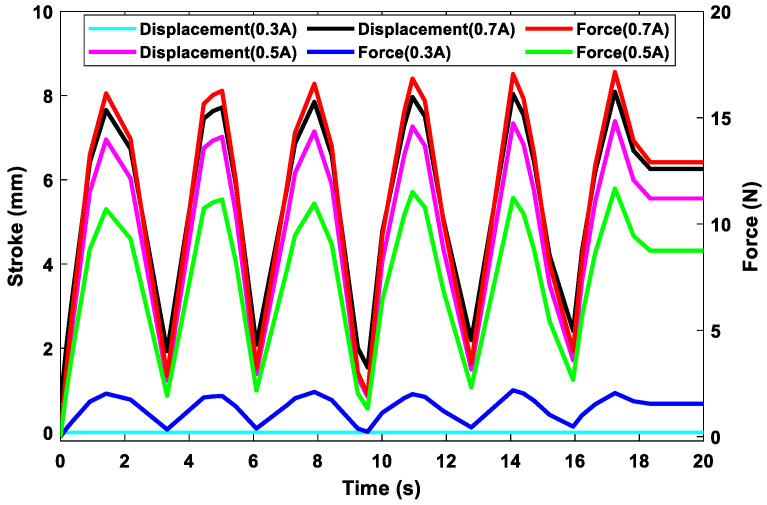
Force generated by the SMA spring and stroke required to open the thermostat.

**Figure 9 sensors-23-03972-f009:**
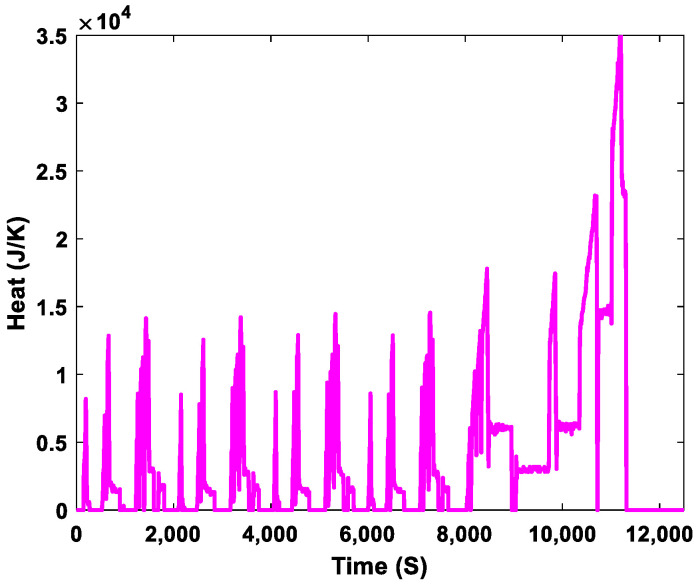
Heat fluctuations in the internal combustion engine.

**Figure 10 sensors-23-03972-f010:**
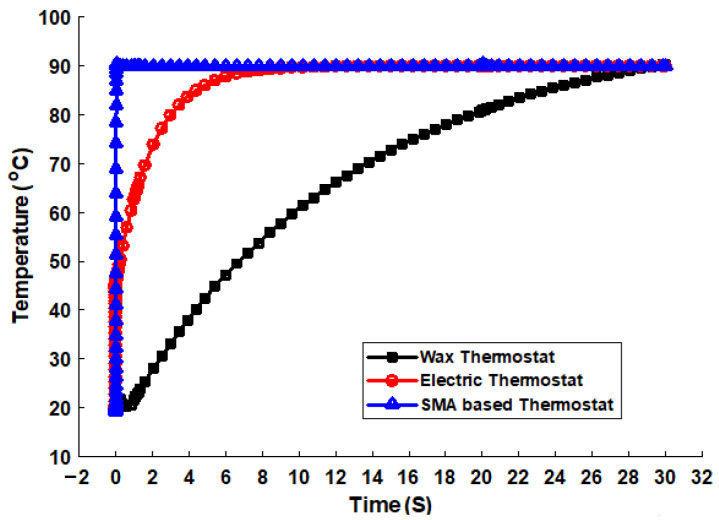
Comparison of the different thermostat types based on their response to the change in the flow direction of the engine coolant.

**Figure 11 sensors-23-03972-f011:**
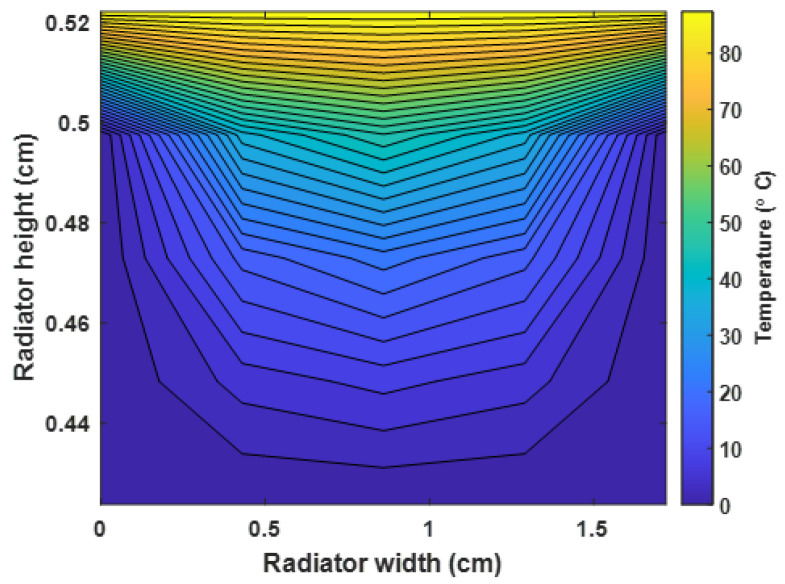
Heat distribution and temperature difference in the radiator after opening the thermostat.

**Figure 12 sensors-23-03972-f012:**
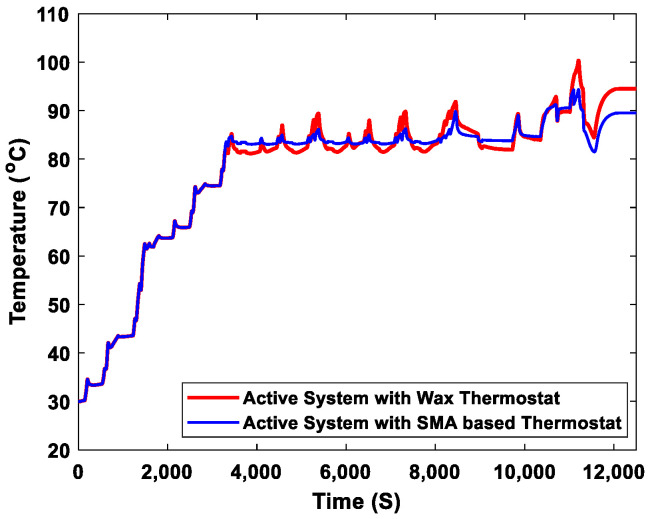
Simulated coolant temperature of the active engine cooling system with a wax thermostat and the active engine cooling system with an SMA-based thermostat.

**Figure 13 sensors-23-03972-f013:**
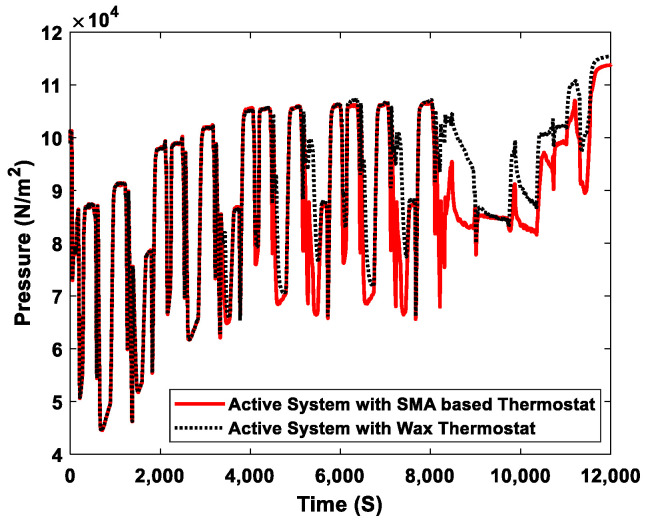
Engine coolant pressure for the active engine cooling system with an SMA-based thermostat and a wax thermostat.

**Table 1 sensors-23-03972-t001:** Design parameters of coiled helical spring.

Symbol	Unit	Meaning
$\tau_{s}$	MPa	Shear stress
G	MPa	Shear modulus
μ	-	Shear strain
*p*	N	Application load
$d_{w}$	mm	Wire diameter
$D_{m}$	mm	Mean coil diameter
$D_{o}$	mm	Outside coil diameter
*L_s_*	mm	Solid length
*L_f_*	mm	Free length
*h*	mm	Pitch
na	-	Number of active coils
*C*	-	Spring index

**Table 2 sensors-23-03972-t002:** Factor used for spring design calculations.

Symbol	Factor	Formula
$C_{s}$	Shear stress correction factor	$\frac{2C\;+\;1}{2C}$

**Table 3 sensors-23-03972-t003:** Material properties of SMA and bias springs.

Property	Unit	Value
SMA Spring (NiTi-Shape Memory Alloy)		
Austenite shear modulus ($G_{a}$)	GPa	70
Martensite shear modulus ($G_{m}$)	GPa	34.5
Austenite shear strain ($\mu_{a}$)	-	0.8%
Martensite shear strain ($\mu_{m}$)	-	0.6%
Martensite electric resistivity	μΩ·cm	80
Austenite electric resistivity	μΩ·cm	100
Austenite temperature ($A_{f}$)	°C	94
Thermal conductivity	W/cm°C	0.18
Young’s modulus (*E*)	GPa	35–70
Tensile strength	GPa	0.8–1
Yield strength	MPa	70–690
Spring index	-	6–10
Poisson’s ratio		0.33
Specific heat	cal/g°C	0.20
Density	g/cm^3^	6.46
Bias Spring (Stainless Steel-AISI Grade 304)		
Density	g/cm^3^	8
Spring index	-	4–12
Specific heat	J/kgK	500
Poisson’s ratio	-	0.27
Yield strength	MPa	205
Tensile strength	MPa	515
Elastic modulus	GPa	193
Electric resistivity	$10^{-9}Ωm$	720
Thermal expansion	$10^{-6}/$°C	17.2
Thermal conductivity	W/m·K	16.2

**Table 4 sensors-23-03972-t004:** Dimensions of the SMA spring and bias spring.

Parameter	Dimension	
	SMA Spring	Bias Spring
$d_{w}$(mm)	1	1.5
$D_{m}$(mm)	7	8.5
L(mm)	242	107
$L_{S}$(mm)	12	6
$L_{f}$(mm)	23	26
P(N)	5.26	17.36
*Mass (g)*	1.23	1.51
$n_{a}$	8	2
$n_{t}$	9	4

## Data Availability

Not applicable.

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
