# Peer review of "A Novel Active Cooling System for Internal Combustion Engine Using Shape Memory Alloy Based Thermostat"

_sensors, 2023, doi:10.3390/s23083972_

Round 1

Reviewer 1 Report

This paper seems to proposed a new thermostat based on SMA for ICE, and some simulations are conducted as they mentioned. However, the paper is very hard to read to get the main methods and findings. I cannot give it a positive recommendation. The following is my main comments.

1. The value of this work is not shown. They give a SMA spring for the valve, but they did not show the importance. The improvement in response time is not a key parameter for the valve.

2. The organization and writing of this paper is disappointing. Many sentences are provided to description the operating of springs, but the information is still vague. The unprofessional words, repeated redundant sentences and grammar errors make the situation harder. A great simplification and better sketch map is needed.

3. Many equations are given in the paper. However, I cannot get a relationship between them and the following results. E.g. The purpose of Eqs. 1-3? How to get the numbers in Table 4 from Eqs. 4-11 as they said? How to match the ones in subsection 2.4 with the following results in Section 3?

4. About the force and displacement values.

4.1 Line 325 says the required force from SMA spring is 17.36N. Why?

4.2 The results in Fig. 7 shows the force is less than 10N when they produce a displacement >35mm. What about the displacement when a 17.36N force is generated?

4.3 How to get the results in Figs. 7&8? What is the relationship between Fig.7 and Fig.8?

4.4 Why only 4 curves are shown in Fig.8 but 6 indicators are given?

4.5 As they said, the force under 0.3A current is very very low. Why?

5. How to get the values in Figures 9-12? What about the information of used modes or objects?

6.A experimental should be fabricated to validate the feasibility of proposed valve. The given simulations are too poor to show the feasibility.  

Reviewer 2 Report

1.     The key words of the article are mainly ‘cooling system’ and ‘SMA thermostat’, please just focus on them. Till now too many redundancies description affect reading, such as too many details about the conventional thermostat.

2.     The design is novel because the materials applied for this situation, however, the structure itself is not novel. Please highlight the progressiveness side and briefly discuss derivation formulas of the traditional mechanisms.

3.     The poor fatigue performance of the SMA is always one limitation for the material, please compare this property with wax thermostat base on the practical demand.  

Reviewer 3 Report

The authors present extensively a novel concept of a thermostat based on the use of SMA.

The introduction provides sufficient elements to have an overall view of the state of the art and a clear vision of the proposed system.

The structure of the paper is consistent and the logic behind is easy to follow, even for a non-expert reader.

There is only a minor remark from my side: in the conclusions, it is stated that the "the thermostat can open the main valve in a time range of milliseconds". However, such a claim cannot be matched elsewhere in the text. Moreover, it is difficult to imagine a "heat-based" system to have a response lower that 1-0.1 sec... and this is instead actually reported, as for instance, lines 586-587: "However, the SMA-based thermostat responds in less than half a second to the opening of the thermostat." May the authors provide some explanation for that, and possibly correct the above-mentioned statements where applicable?

Reviewer 4 Report

It is a correct technological work about the design of an active cooling system for an internal combustion engine by using a thermostat based in a shape memory alloy.  I recommend a major revision and, regarding the topic, to resubmit to other mdpi journal.

The design of this active cooling system has been performed via COMSOL Multiphysics and MATLAB software. Thus, there are simulated conditions. There are several technological issues linked to the fabrication of this system and the working conditions that has been not taken into account.

My point of view is that is not a manuscript based on sensors. I recommend to resubmit to other mdpi journal. One option is Technologies.

a)       The authors should remark the key factors associated to the future fabrication of this device.

b)      The authors should improve the manuscript discussing the future working conditions: stability of the thermo-mechanical response under long-term cycling experiments, cracks, fatigue, wear, ...

c)       There are some sentences based on the simulation, but “dangerous” in working conditions. “stable fluctuation pressure in the upper radiation dose”,  “engine components with less wear”.

Round 2

Reviewer 1 Report

The revised version solved many issues, and the paper has been greatly improved. 

Reviewer 4 Report

The authors modify the article taking into account the comments of the referees.

The quality and soundness of the manuscript has been improved.